# “Lived the Pandemic Twice”: A Scoping Review of the Unequal Impact of the COVID-19 Pandemic on Asylum Seekers and Undocumented Migrants

**DOI:** 10.3390/ijerph19116624

**Published:** 2022-05-29

**Authors:** Zelalem Mengesha, Esther Alloun, Danielle Weber, Mitchell Smith, Patrick Harris

**Affiliations:** 1Centre for Health Equity Training, Research & Evaluation (CHETRE), UNSW Australia Research Centre for Primary Health Care & Equity, A Unit of Population Health, Member of the Ingham Institute, Sydney, NSW 2170, Australia; patrick.harris@unsw.edu.au; 2Health Equity Research and Development Unit (HERDU), UNSW Australia Research Centre for Primary Health Care & Equity, Sydney Local Health District, Sydney, NSW 2050, Australia; e.alloun@unsw.edu.au; 3NSW Refugee Health Service, South Western Sydney Local Health District, Sydney, NSW 2170, Australia; danielle.weber@health.nsw.gov.au (D.W.); mitchell.smith1@health.nsw.gov.au (M.S.)

**Keywords:** COVID-19, asylum seekers, undocumented migrants, impact, inequity

## Abstract

Background: Emerging evidence suggests that the COVID-19 pandemic is widening pre-pandemic health, social, and economic inequalities between refugees, migrants, and asylum seekers and the general population. This global scoping review examined the impact of the pandemic on community-based asylum seekers and undocumented migrants in high- and upper-middle-income countries. Methods: We conducted a systematic search of peer-reviewed articles in *PubMed*, *Scopus*, *Web of Science*, and *ProQuest Central*. We applied Katikireddi’s framework of understanding and addressing inequalities to examine the differential impact of the pandemic across exposure, vulnerability to infection, disease consequences, social consequences, effectiveness of control measures, and adverse consequences of control measures. Results: We included 32 articles in the review. The analysis showed that asylum seekers and undocumented migrants experienced greater exposure to the COVID-19 virus and higher infection rates. They also experienced differential social consequences in the form of job loss and lost and/or reduced work hours. The effectiveness of pandemic response measures on asylum seekers and undocumented migrants was also affected by pre-pandemic social and economic marginalisation, exclusion from pandemic-induced policy measures, lack of appropriate pandemic communication, and variable trust in governments and authority. Pandemic control measures had greater adverse consequences on asylum seekers and undocumented migrants than the general population, with the majority of studies included in this review reporting worsened mental health and social isolation conditions and reduced access to health care. Conclusions: Asylum seekers and undocumented migrants experienced a disproportionate impact of the COVID-19 pandemic across the six thematic areas of comparison. Policies that reduce exposure and vulnerability to the infection, grant equitable access to health and social care, and build capacities and resilience, are critical to enable asylum seekers and undocumented migrants to cope with and recover from pre-pandemic and pandemic-induced inequalities.

## 1. Introduction

The COVID-19 pandemic has infected nearly 500 million people and has caused over 6 million deaths globally at the time of writing [1]. It has also contributed to negative health and socioeconomic outcomes worldwide, especially to marginalised groups such as refugees, migrants, and asylum seekers [2]. Refugees experience significant economic hardship due to disproportionate job loss, difficulty accessing relief, and reduced support during COVID-19 lockdown measures [3]. Lack of linguistically appropriate pandemic information for refugees and barriers to effective virtual communication hampered the success of public health control and economic recovery efforts [4]. There is also evidence suggesting that refugees faced reduced volunteer and public services during the pandemic as volunteers and staff are restricted by government mandates, which disturbs provision of resettlement support services [5]. This evidence implies that the impact of COVID-19 may further widen the pre-pandemic social, health, and economic gaps between refugees and host populations in countries of resettlement.

Evidence suggests that refugee and migrant populations are not homogeneous in relation to their health care needs and experiences during the COVID-19 pandemic [6]. Specifically, those without legal status and asylum seekers experience greater health and social inequalities compared to officially resettled refugees [7]. Lack of health care rights in countries of asylum, living in humanitarian shelters, and lack of equitable access to research participation, particularly for conditions that disproportionately affect them, such as COVID-19, are among the factors that contribute to these variations [8,9]. Additionally, public policies prepared in response to the COVID-19 pandemic largely overlooked the needs of community-based asylum seekers and undocumented migrants [10]. Policies that considered the health needs of these populations have major gaps; health care entitlements are not made explicit, and diverse characteristics such as language, age, and gender not adequately considered [11]. The invisibility of asylum seekers and undocumented migrants in health policies negatively impacted COVID-19 control efforts, including access to vaccines [12].

In this review, we focus on two highly vulnerable groups of migrants: community-based asylum seekers, i.e., those who are living in the community with the host population while awaiting their refugee determination process, and undocumented migrants. The United Nations High Commissioner for Refugees (UNHCR) identifies an asylum seeker as a person who has left their country and is seeking protection from persecution and serious human rights violations in another country but whose request for sanctuary has yet to be processed [13]. Undocumented migrants are people who are living in another country whom the government does not consider to have the legal right to remain [14]. Emerging evidence from systematic reviews revealed that migrants and forcibly displaced populations experienced higher rates of COVID-19 outbreaks [15] and negative mental health impacts because of lockdown measures [16]. However, there is limited evidence on the risk of exposure, vulnerability to infection, COVID-19 infection consequences, social consequences, and adverse impacts of control measures among asylum seekers and undocumented migrants living in high- and middle-income countries. These individuals frequently do not have health and social security rights in their countries of residence because of their legal and/or visa status. Therefore, in this review, we sought to examine the unequal impact of the COVID-19 pandemic using Katikireddi and colleagues’ [17] framework of understanding and addressing inequalities. The framework examines inequalities under the following themes: differential exposure, differential vulnerability to infection/disease, differential disease consequences, differential social consequences, differential effectiveness of control measures, and differential adverse consequences of control measures. The framework is a useful heuristic to illuminate the different pathways that have generated inequities during the pandemic and their complex interactions. This is important to gain a fuller understanding of the multiple dimensions that may give rise to suboptimal health and social outcomes for asylum seekers and undocumented migrants [17].

Partly related questions have been raised by other researchers, but past reviews have focused on clinical outcomes [18], assessing the impact of the COVID-19 pandemic on officially resettled refugees and asylum seekers [15] and all ethnic and racial minorities [19]. Furthermore, these reviews were conducted in the early stages of the pandemic and showed a limited picture of the impact on asylum seekers and undocumented migrants [15]. This research provides a review of the latest peer-reviewed literature with a focus on the impact of COVID-19 on community-based asylum seekers and undocumented migrants. These results will have the potential to inform future research and pandemic response measures targeting these population groups.

## 2. Methods

The search strategy followed the Preferred Reporting Items for Systematic Reviews and Meta-Analyses (PRISMA) guidelines [20]. The following databases were searched for studies published between December 2019 and July 2021: *PubMed*, *Scopus*, *Web of Science*, and *ProQuest Central*. A set of search terms (Table 1) used for each area of interest was compiled. The database search results were imported into a single library in EndNote (Clarivate Analytics, USA) where duplicates were removed. The combined library was imported into Covidence systematic review software (Veritas Health Information, Australia) for title/abstract and full-text screening.

### 2.1. Inclusion and Exclusion Criteria 

Articles were included in the review if they: (i) examined the perspectives of community-based asylum seekers and undocumented migrants about COVID-19, including its impact; (ii) contained empirical research; (iii) were conducted in high and upper-middle-income countries [21]; and (iv) were published in peer-reviewed journals. Articles were excluded if they: (i) were commentaries, reviews, letters, books or grey literature; (ii) were modelling studies; (iii) focused on permanently and/or officially resettled refugees and migrants; (iv) did not contain data related to pandemic-induced impacts and measures; or (v) were not related to the COVID-19 pandemic. The PRISMA flow diagram (Figure 1) provides reasons for exclusion. 

### 2.2. Study Selection

Using the inclusion and exclusion criteria, titles and abstracts of all articles retrieved were assessed by two independent reviewers (Z.M. and E.A.), and a 20 per cent sample was reviewed by a third reviewer (D.W.) to address risk of selection bias. Where it was unclear whether the selection criteria were met, studies were included for full-text review. All full-text articles were reviewed by two independent reviewers (Z.M. and E.A.). Disagreements were resolved by a third reviewer. 

### 2.3. Data Analysis and Synthesis

The analysis was informed by a framework developed by Katikireddi and colleagues [17], which examines the unequal impact of the pandemic across six pathways. A deductive thematic narrative analysis [22] was performed on each paper identified for inclusion in the final synthesis, whereby any references (qualitative or quantitative) to COVID-19 and its impact on health, social, and economic outcomes from the perspectives of asylum seekers and undocumented migrants were coded using NVivo qualitative data analysis software (QSR International Pty Ltd. (Burlington, MA, USA) Version 12, 2018). Only direct quotes or statistical data from included studies were analysed. Interpretive or synthesised data within papers were excluded. Where studies included perspectives from officially resettled refugees and migrants, these data were excluded from the analysis. Deductive coding using the framework enabled the emergence of key concepts, which are presented in the results. 

### 2.4. Patient and Public Involvement

We did not involve patients and the public in the development of the research questions and conduct of this scoping review. 

### 2.5. Ethics Approval 

We used publicly available data for this review, and ethics approval was not required. 

## 3. Results 

The database search identified 12437 potential studies. After removal of duplicates, 5490 titles and abstracts were screened. Of these, 190 full-text publications were retrieved for consideration. A total of 158 articles were excluded after performing the full-text review, leaving 32 articles for inclusion (Figure 1). The characteristics of the 32 included articles are summarised in Table 2. Two-thirds (*n* = 29) of the studies were published in 2021. The majority of studies were conducted in Europe (12) and North America (11), with a smaller number conducted in South America (2). Quantitative methodologies were used by the majority of studies (18), with the remainder involving qualitative (10) and mixed methods (4). We synthesised the findings from the review against Katikireddi et al.’s [17] framework. 

**Table 2 ijerph-19-06624-t002:** Summary of included articles.

First Author	Year	Country of Study	Population	Ethnicity	Methodology	Data Collection Approach	Main Framework Pathway/s Captured and Outcome Reported
Al-Awaida [23]	2021	Jordan	Refugees seeking asylum	Syrian	Quantitative	Survey (*n* = 2380)	Differential adverse consequences of control measures. Demonstrated a high prevalence of PTSD, in which PTSD showed incidence of 82.5% and 66.5% in Syrian refugees and Jordanian populations, respectively.
Alsharif [24]	2021	Saudi Arabia	Undocumented migrants	Sub-Saharan Africa and Southeast Asia	Qualitative	Interview (*n* = 15)	Differential disease consequences and differential effectiveness of control measures. Undocumented migrants do not access health care due to fear of deportation.
Aragona [25]	2021	Italy	Asylum seekers, refugees, and forced and undocumented migrants.	Africa, Europe, Middle East, South/Central America, and Asia	Quantitative	Survey (*n* = 81)	Differential adverse consequences of control measures. Mental health treatment adherence negatively impacted by COVID-19, 32% discontinued pharmacological treatment, and 52% discontinued psychotherapy.
Aragona [26]	2020	Italy	Asylum seekers, refugees, forced migrants, undocumented migrants	Africa, Europe, Middle East, South/Central America, and Asia	Quantitative	Electronic medical record data (*n* = 555)	Differential adverse consequences of control measures. Mental health follow-up treatment attendance shows higher decline compared to previous years (30% in 2020 vs. 17% in 2017–2019).
Aung [27]	2021	Malaysia	Refugees seeking asylum	Rohingya	Quantitative	Survey (*n* = 283)	Differential vulnerability to infection. Demonstrated high health and social vulnerabilities for the COVID-19 infection.
Baggio [28]	2021	Switzerland	Undocumented migrants	N/A	Quantitative	Survey (*n* = 215)	Differential vulnerability to infection and differential disease consequences. Proportion of positive tests significantly higher among undocumented migrants (32.1% vs. 23.6%) compared to host population.
Banati [29]	2020	Lebanon	Refugees seeking asylum	Syrian and Palestinian	Qualitative	Interview (*n* = 100)	Differential adverse consequences of control measures. COVID-19 compounds pre-existing disadvantage: issues in getting food and supplies, intra-family problems, fear of violence and scapegoating, anxiety about the future, social isolation, lack of privacy worsening.
Bigelow [30]	2021	United States	Undocumented migrants	Latinx	Quantitative	Electronic medical record (*n* = 1786)	Differential exposure and differential disease consequences. Highest positivity rate detected among Latinx at (31.5%) 10 times higher than whites.
Blackburn [31]	2021	United States	Undocumented migrants and service providers	Hispanic	Qualitative	Interview (*n* = 13)	Differential adverse consequences of control measures. Anti-immigrant rhetoric has made undocumented migrants less willing to access healthcare.
Budak [32]	2020	Turkey	Asylum seekers	Syrian	Quantitative	Survey (*n* = 414)	Differential disease consequences. Some groups underestimate seriousness of COVID-19; not enough information or PPE is available.
Burton [7]	2020	Switzerland	Undocumented migrants	Latin American, Asia, Africa, and Non-EU Europe	Mixed methods	Survey (*n* =117) and interview (*n* = 17)	Differential disease consequences, differential social consequences and differential adverse consequences of control measures. Identified high prevalence of exposure to COVID-19, poor mental health along with frequent avoidance of health care, and loss of work and income.
Cervantes [33]	2021	United States	Undocumented migrants	Latinx	Qualitative	Interview (*n* = 60)	Differential exposure, differential social consequences, and differential effectiveness of control measures. Patients who survived hospitalisation described initial disease misinformation and economic and immigration fears as having driven exposure and delays in presentation.
Deal [34]	2021	United Kingdom	Asylum seekers and undocumented migrants	Africa, Venezuela, Eastern Mediterranean and Europe, and Sri Lanka	Qualitative	Interview (*n* = 32)	Differential exposure, differential social consequences, and differential effectiveness of control measures. Majority are hesitant in accepting vaccines and facing multiple unique barriers to access (lack of accessible information and poor health literacy, fear of deportation, distrust).
Devillanova [35]	2020	Italy	Undocumented migrants	*n*/A	Quantitative	Survey (*n* = 1590)	Differential exposure, differential disease consequences, and differential effectiveness of control measures. Lockdown triggered sharp reduction in health visits, increased number of presentations with COVID-19 symptoms, shutdown of outpatient clinics, and patients reporting deteriorating housing conditions.
Fiorini [36]	2020	Italy	Undocumented migrants	Africa, Asia, Latin America, Eastern Europe	Quantitative	Survey (*n* = 272)	Differential exposure, differential vulnerability to infection, and differential disease consequences. All had risk factors and predispositions that increased severity and outcomes.
Gosselin [37]	2021	France	Undocumented migrants and asylum seekers	Sub-Saharan Africa	Quantitative	Survey (*n* = 100)	Differential effectiveness of control measures and differential adverse consequences of control measures. Food insecurity was more often reported during lockdown than before (62% vs. 52%) and increased rate of severe depression post lockdown.
Hajjar [38]	2021	Lebanon	Refugees seeking asylum	Syrian	Quantitative	Survey (*n* = 129)	Differential exposure and differential adverse consequences of control measures. Documented massive job loss and reduced wages, discontinued education for children, and high stress and anxiety due to lack of assistance.
Hamadneh [39]	2021	Jordan	Refugees seeking asylum	Syrian	Quantitative	Survey (*n* = 389)	Differential effectiveness of control measures. Refugee mothers were knowledgeable about COVID-19 transmission and prevention but lacked knowledge about transmission between mother and child and smoking risks associated with COVID-19.
Karajerjian [40]	2021	Lebanon	Refugees seeking asylum	Syrian	Qualitative	FGD and interview (*n* = 50)	Differential exposure and differential adverse consequences of control measures. COVID-19 compounds disadvantage and mental health issues for refugee women already in precarious situations. Fear and anxiety about the disease is high, and access to healthcare is uncertain.
Kondilis [41]	2021	Greece	Asylum seekers and refugees	N/A	Quantitative	Retrospective surveillance data	Differential disease consequences. Twenty-five COVID-19 outbreaks were identified in refugee and asylum-seeker reception facilities.
Longchamps [42]	2021	France	Undocumented migrants	Europe, Africa, Eastern Mediterranean	Quantitative	Survey (*n* = 240)	Differential vulnerability to infection and differential effectiveness of control measures. Reported significant vaccine hesitancy (41%).
MacCarthy [43]	2020	United States	Undocumented migrants, asylum and humanitarian visa holders	Latinx	Mixed methods	Survey and interview (*n* = 52)	Differential adverse consequences of control measures Participants reported increased interpersonal conflict and alcohol consumption due to lockdown; disruption in accessing medical care, job loss, and no assistance for those undocumented.
Martuscelli [44]	2020	Brazil	Refugees seeking asylum	Syria, DRC, Guyana, Venezuela	Qualitative	Interview (*n* = 29)	Differential adverse consequences of control measures. Refugees are affected by border closures and their rights to documentation, healthcare, and social assistance (state emergency benefit) are violated.
Martuscelli [45]	2021	Brazil	Refugees seeking asylum	Syria, DRC, Guyana, Venezuela	Qualitative	Interview (*n* = 29)	Differential exposure, differential effectiveness of control measures, and differential adverse consequences of control measures. Continued lack of culturally and linguistically adapted information makes accessing services difficult.
Quandt [46]	2020	United States	Undocumented migrants, mixed-status families, and residents	Latinx	Mixed methods	Survey (*n* = 105)	Differential exposure, differential effectiveness of control measures, and differential adverse consequences of control measures. Families engaged in frequent interpersonal contact that could expose community members and themselves to COVID-19.
Quandt [47]	2021	United States	Undocumented migrants, mixed-status families, and residents	Latinx	Mixed methods	Survey (*n* =105)	Differential exposure, differential effectiveness of control measures, and differential adverse consequences of control measures. Rural workers reported fewer workplace protective measures for COVID-19. Fear and anxiety, particularly about finances and children, dominated their experiences.
Redditt [48]	2020	Canada	Asylum seekers	N/A	Quantitative	COVID-19 outbreak management data (*n* = 60)	Differential effectiveness of control measures and differential disease consequences. Documented a very high rate of infection in a humanitarian shelter (41.7% of tested residents are positive).
Reynolds [49]	2021	Mexico	Asylum seekers and service provides	El Salvador, Nicaragua, Honduras, Mexico, Cuba, Bolivia, and Guatemala	Qualitative	Interview (*n* = 30)	Differential effectiveness of control measures and differential adverse consequences of control measures. COVID-19 caused mental health burdens and less adherence to disease-reduction strategies. Control measures created distrust and decreased health care services use.
Sabri [50]	2021	United States	Undocumented immigrants and service providers	Asian, Latina, African	Qualitative	Interview (*n* = 62)	Differential adverse consequences of control measures. COVID-19 is connected to increased intimate partner violence, and assistance is not available to undocumented women.
Serafini [51]	2021	United States	Undocumented migrants	Hispanic	Quantitative	Survey (*n* = 35)	Differential adverse consequences of control measures. Participants reported worsened anxiety (49%) and depression (46%) levels due to the pandemic.
Terp [52]	2021	United States	Undocumented migrants	N/A	Quantitative	Review of death reports (*n* = 35)	Differential vulnerability to infection and differential disease consequences. COVID-19 is a leading cause of death rate among undocumented migrants.
Turunen [53]	2021	Finland	Asylum seekers	N/A	Quantitative	Screening tool + medical record review and interviews (*n* = 260)	Differential exposure and differential disease consequences. High COVID-19 infection rate identified among asylum seekers.

### 3.1. Differential Exposure 

Several studies identified that asylum seekers and undocumented migrants experienced increased exposure to the SARS-CoV-2 virus due to living in overcrowded housing [33,35,38,40,45,46] and shared accommodation [34,36]. This made it difficult to adopt COVID-19 public health measures (social distancing, quarantine, and self-isolation) and infection concerns were high: “if someone is infected all the people in the house will die” [33]. Many asylum seekers and undocumented migrants continued to work during the peak of the pandemic in conditions where it was difficult to follow control measures [33,46,47]. Asylum seekers represented in these studies explained that “other people can work from home but our jobs don’t allow us to” [47] and “our work requires heavy lifting and we are breathing hard, and the mask doesn’t help” [33]. We identified evidence suggesting increased exposure led to higher infection rates among asylum seekers. For instance, undocumented migrants in the U.S. who tested positive were significantly more likely to be from a larger household (*p* < 0.002) [30]. One-third of asylum seekers living in a reception centre in Finland tested positive for COVID-19 [53]. 

### 3.2. Differential Vulnerability to Infection

Once exposed to the virus, “vulnerability” factors determine the chance of developing the disease. We identified differences between community and hospital-based studies regarding variations in pre-existing comorbidities. Based on a community-based sample of participants, three studies reported no difference in the proportion of participants having comorbidities between undocumented migrants and the general population [27,28,42]. However, one record-based study reported that all undocumented migrants attending a clinic due to symptoms had chronic comorbidities that worsened the severity and outcome of the infection [36]. All undocumented migrants who died of COVID-19 in U.S. immigration detention centres had at least one chronic disease condition [52]. 

### 3.3. Differential Disease Consequences 

Asylum seekers and undocumented migrants avoided seeking health care during the pandemic due to lack of trust in the system and fear of jail and deportation [7,24,33], which made it difficult to determine the exact burden of the COVID-19 pandemic case numbers, admissions, and deaths. However, multiple studies that examined differential disease consequences reported a disproportionately higher proportion of COVID-19 cases among asylum seekers and undocumented migrants compared to the general population [30,32,35,36,41,48]. For instance, the positivity rate at a community testing centre in the U.S. was 10 times higher among undocumented migrants than among non-Hispanic whites (91.6% vs. 81.7%, *p* < 0.001) [30]. Voluntary mass screening at reception centres for asylum seekers in Finland [53] and Greece [41] identified multiple COVID-19 outbreaks. Undocumented migrants and homeless persons in Geneva experienced a higher positivity rate for COVID-19 compared to the general population (32.1% vs. 23.6%, *p* = 0.005) [28]. In another study, although undocumented migrants with the infection had a comparable proportion of hospitalisations with other population groups [28], a higher proportion required ICU treatment [33] and complications attributed to the majority (74.2%) of deaths occurring among undocumented migrants in the U.S. [52]. 

### 3.4. Differential Social Consequences of the Disease

Studies that examined the social consequences of the disease on infected asylum seekers and undocumented migrants noted that it led to financial loss including job loss [33] and loss of work hours [7]. This is in part because asylum seekers and undocumented people tend to be precariously employed and in low-skill work that made them “dispensable” during the pandemic [33]. Considering the lack of social safety nets as undocumented people, infection with COVID-19 also increased the risk of homelessness [7,33], sometimes driving further exposure due to moving in with friends and families in overcrowded housing [33]. Food insecurity was another negative repercussion of infection among undocumented migrants [7,33].

### 3.5. Differential Effectiveness of Control Measures

Studies examined for this review showed that public health measures introduced to control the pandemic were not equally effective across population groups. We grouped control measures into two categories: first, information and communication mediated by trust in institutions; second, testing and vaccination.

#### 3.5.1. Information and Communication

Several studies documented low levels of knowledge about transmission and symptoms of COVID-19 [37,39,54] and poor understanding of infection control measures [49] compared to the host population. Several factors contributed to differential effectiveness of information and communication interventions. On the one hand, studies noted a lack of culturally and linguistically appropriate information in the U.S. [33,46], U.K. [34], Saudi Arabia [24], and Brazil [45] for asylum seekers and undocumented migrants. On the other hand, barriers to information were infrastructural (rather than linguistic), with a lack access to official media channels leading to misinformation about the pandemic [33]. 

In addition, communication of government information specifically targeting groups with precarious or no legal status was not effective: none of the undocumented and asylum seekers interviewed in a U.K. study had heard of recent announcements that vaccines would be given without immigration checks [34]. In France, only a few knew of the extension of residency permits and State Medical Aid [37]. In both countries, the evidence suggested that government information was insufficient [37] and generated “a lot of confusion” [34]. Consequently, asylum seekers felt “most abandoned or scared due to a lack of understandable, clear official information” and relied on informal and unreliable information sources [34], which had implications for vaccine and/or testing uptake [34,42].

Lastly and crucially, lack of trust in government and authorities is a pervasive problem that impacts the effectiveness of control measures [55]. Asylum seekers and undocumented people’s experience of this distrust is unique given they are at the coalface of states’ securitised border control, and past or current institutional violence [24,34,49]. As such, most participants in a U.K. study expressed suspicion towards the government’s COVID-19 response measures: “I don’t know to what extent it is true. It might be a ploy to get people to come [and detain them]” as well as scepticism: “we asylum seekers will come in the end [about the vaccine]”; “they never care for the less privileged” [34]. A similar phenomenon was observed among undocumented migrants in Saudi Arabia [24] where most undocumented people did not trust nor intend to make use of the government’s amnesty granting them access to medical services. Asylum seekers at the U.S.–Mexico border also showed distrust for official institutions and non-governmental organizations (NGOs) and therefore did not come forward for healthcare or testing during the pandemic [49].

#### 3.5.2. Vaccination

Several equity-related issues emerged relevant to vaccination access, availability, and appropriateness, with adverse consequences for asylum seekers and undocumented people. Vaccine availability was uneven in some settings and concerns for safety, living conditions, and resettlement prevailed over vaccination: “even if they vaccinate us, and we continue to live in these conditions, what’s the point?” [47] Vaccine hesitancy among undocumented migrants in a study in France was equivalent to that of the general French population but with no legal residence (OR = 0.51, 95% CI 0.27–0.92) and poor health literacy (OR = 0.38, 95% CI 0.21, 0.68) being associated factors [42]. This echoes the situation in the U.K., where information and trust deficits also drove significant hesitancy among asylum seekers and undocumented people [34]. This group faced significant barriers to access, such as direct and indirect costs, fear of immigration checks, and pre-existing issues accessing mainstream primary care, such as language barriers and the registration process for an appointment with a GP [34].

### 3.6. Differential Adverse Consequences of Control Measures 

Adverse psycho-social consequences of control measures were wide-ranging and widespread in the studies we reviewed. Different causal pathways emerged at the individual, community, and societal levels, invoking social determinants of health that spanned access to healthcare and support, economic stability (employment, food, and housing security), and social isolation. In this context, asylum seekers and undocumented migrants faced unique and disproportionate negative consequences, which we elaborate on in this section.

#### 3.6.1. Differential Mental Health Consequences

The impacts of COVID-19 control measures had a damaging effect on mental health as documented in twenty-one studies included in this review. A study six weeks into the Swiss lockdown showed that anxiety and depression were more prevalent among undocumented people compared to recently regularised people (71.4% vs. 66.7%) [7]. The deterioration in mental health associated with pandemic control measures was also worse for asylum seekers and undocumented migrants in Jordan (*p* < 0.05) [23], Mexico [49], and Lebanon [38] compared to other groups. Qualitative evidence paints an equally dire picture of complex mental health issues in very varied settings, with some COVID-specific fears among undocumented migrants in precarious jobs and housing [47]. However, there was also widespread tension, anxiety, and uncertainty brought on by the pandemic and restrictions for asylum seekers and undocumented migrants [29,33,40,44,47,49,50] on top of existing instability: “We are heading towards an unknown path, we cannot turn back and we will die in both cases either from the virus or from hunger, if the lockdown continues” [29]. There are also some studies that showed the direct impact of lockdown on rising depression and anxiety by comparing before and after lockdown rates. In one French study, the prevalence of depression increased (72% vs. 65%) [37]. Symptoms also worsened among undocumented migrants in Italy (50% worsened anxiety, 38% worsened depression) [25] and the U.S. (49% worsened anxiety, 46% worsened depression) [51]. 

#### 3.6.2. Restricted Access to Health Care

The mental health of asylum seekers and undocumented migrants was further compromised by reduced access and use of health services during the pandemic. For instance, fewer undocumented migrants visited a free psychiatric clinic for migrants in Italy, with only 17.5% returning for follow-up treatment during the peak of the pandemic [26] and 32% discontinuing psychopharmacological treatment [25]. A similar pattern was observed in another general Italian clinic, with women experiencing the sharpest decline (77%) in visits [35]. Elsewhere, the pandemic stretched the limited available medical resources even further, with asylum seekers facing additional barriers to access non-COVID care [31,40,49]. In the U.K., discontinuation and/or digitalisation of services meant asylum seekers and undocumented people struggled to access care [34].

#### 3.6.3. Exclusion from Pandemic Induced Policy Measures

Some studies included in this review highlighted that asylum seekers and undocumented migrants were routinely excluded from social benefits [33,35,46]. They were also specifically excluded from unemployment and pandemic stimulus policy measures and benefits in the U.S. [43,47], not informed about their eligibility in Brazil [45], and disincentivised from accessing them in Switzerland since financial autonomy remained a condition for resident applications [7]. Availability of other socio-economic support was also curtailed by the pandemic: three studies in Lebanon reported a shortfall of humanitarian assistance [29,38,40], while many dedicated NGO social services closed [7,34,50]. 

#### 3.6.4. Visa Processing Delays

While border closures and delays with visa processing affected many people with transnational ties and families, it posed unique challenges for asylum seekers and undocumented people who were already vulnerable and in precarious situations. For instance, in Brazil, all asylum, naturalisation, and family reunification procedures were suspended with the closure of the government agency in charge, leaving people in limbo, without status or a timeline [44]. Undocumented migrants in the U.S. [50] or those seeking entry there [49] also reported delays in asylum determination and longer family separation. 

#### 3.6.5. Economic Instability

Economic instability from loss of work with flow-on consequences of food and housing insecurity was “by far the largest impact” of government-imposed curfews, lockdowns, and travel restrictions on asylum seekers and undocumented migrants [25,49]. Asylum seekers and undocumented migrants often work in the informal sector [45] and rely on work in people’s homes [7], all sectors and services that were heavily impacted by pandemic control measures. The pandemic also eroded solidarity and support networks as time passed under lockdown, and the economic situation worsened: “None of my friends can lend me money as they are all facing difficulties” [7]. As a result, the loss of employment and financial pressure took a heavy toll with rising cost of living and inability to buy essential items including food [51]. For instance, undocumented migrants in Switzerland reported more food insecurity (77.4% vs. 54.1%) and food insecurity with hunger (29.9% vs. 20%) during lockdown than regularised people (*p* = 0.025) [7]. While food insecurity among undocumented migrants was more often reported during lockdown than before in France [37] and the U.S. [43,47,51], the atmosphere in some settings was that “dying from the virus is better than dying from hunger” [29]. The COVID-19 pandemic control measures also exacerbated the pre-pandemic housing insecurity among asylum seekers and undocumented migrants [7,51] with people unable to pay rent [40,43,45] and/or facing eviction [29]. In one Italian study, the share of homeless undocumented migrants nearly doubled in the post-lockdown period [35]. 

#### 3.6.6. Social Isolation 

There is evidence that the general population experienced widespread social isolation due to pandemic control lockdown and social distancing measures [56]. Yet, asylum seekers and undocumented migrants were uniquely and more adversely impacted because they have had to move countries and were already uprooted from their communities and social networks through forced migration [29,45]. Exile and the lack of formal status already create conditions of invisibility and social isolation [7,45]. The added insecurity and instability created by the pandemic combined with physical distancing, stay at home orders, and fear of infection generated even more social separation, loneliness, and isolation from their support circles [29,33,40,43,47,49,50], with women and girls [29,40,50] singled out as particularly badly affected. Fresh stigmatisation [29], xenophobia, and discrimination [31,45,47,50] fuelled by the pandemic and targeting asylum seekers and undocumented migrants also contributed to increased rates of social isolation among these groups. 

It has also been said asylum seekers and undocumented migrants “lived the pandemic twice” by experiencing the disproportionate effects of the pandemic in their countries of residence and worrying about pandemic impacts on their families in their countries of origin [45]. 

## 4. Discussion

This review examined the impact of the COVID-19 pandemic on asylum seekers and undocumented migrants—already some of the most marginalised people in the global community prior to the pandemic. The available studies indicate that asylum seekers and undocumented migrants experienced greater exposure to the virus, higher COVID-19 infection rates, and differential social consequences in the form of job loss and lost or reduced work hours. Measures introduced to curb the spread of the pandemic were not equally effective, with asylum seekers and undocumented migrants experiencing a lack of culturally and linguistically appropriate information. The low level of awareness of pandemic control measures and policies and lack of trust in government and authorities impacted the effectiveness of control measures. COVID-19 control measures also had greater adverse consequences on asylum seekers and undocumented migrants, with the majority of studies included in this review reporting worsened mental health conditions compared to pre-pandemic levels and to non-migrant populations. Although asylum seekers and undocumented migrants had comparable COVID-19 vaccine acceptability rates, barriers such as fear of immigration checks, language, and previous negative experiences in accessing health care hampered vaccine uptakes. These findings indicate that the COVID-19 pandemic exacerbated pre-pandemic inequities between undocumented migrants and asylum seekers and the general population. In some cases, this has led to calls for public enquiries into the handling of the pandemic and its impacts on vulnerable groups such as asylum seekers and precarious migrants [57]. 

The studies we analysed illuminate evidence of significant adverse health and social consequences for this group during the pandemic. Our review further shows that many of the inequalities documented in the literature are systematic and avoidable in that they are due, at least in part, to policy decisions. Existing policy approaches that restrict access to healthcare and social security benefits were compounded by pandemic response measures that neglected the health and social care needs of asylum seekers and undocumented migrants in their diversity, varied legal status, and intersectional social location. In some cases, undocumented or recently regularised migrants were excluded entirely from COVID-19 policy responses [58], deepening inequities. As such, our results emphasise the importance of including marginalised groups in health and social care systems and embedding universal coverage into public health responses to emergencies such as a pandemic. Facilitating timely granting of visas to confer stable residency is one measure that would improve service access by removing the fear of immigration checks or queries around legal status and service eligibility. Community-based testing and surveillance can help understand the epidemiology of the infection and guide response measures. When developed and delivered with asylum seekers and precarious migrants themselves, testing initiatives can be even more effective [30,58]. 

The results highlight that effective engagement and genuine collaboration with diverse marginalised communities is a key recommendation when designing and implementing pandemic-response policies. Actively involving asylum seekers and other precarious migrants and consulting with frontline providers and advocates (including relevant NGOs, community and faith leaders, and ethno-specific community groups) can help tailor and disseminate health messages in linguistically appropriate and culturally responsive ways. Mobilising existing relationships of trust and care and using established channels reduces fear, misinformation, and mistrust of authorities. This may increase health service use and the uptake of preventive measures, including vaccination. With adequate support, funding, and capacity-building, migrant-led organisations and specialised community services can help bridge some of the gaps identified in this review and increase the general effectiveness of control measures. The broad and specific measures mentioned can improve the lives of asylum seekers and undocumented migrants but also help to prevent and contain outbreaks and adverse disease consequences for individuals and the general community.

### Strengths and Limitations

To the best of our knowledge, this is the first review to examine the disproportionate impact of the COVID-19 pandemic on some of the most marginalised people in the community—asylum seekers and undocumented migrants. The use of Katikireddi and colleagues’ framework allowed us to examine pathways that generated the unequal COVID-19 pandemic effects and identify targets for policy interventions for these target groups.

We included only articles written in the English language and, as a result, may have missed certain other publications. The heterogenous nature of the included studies and population means that meta-analysis of the individual study findings was not performed. The definitions of an asylum seeker or undocumented migrant vary across countries, and in some instances, screening papers that only included these population groups was difficult, as primary studies do not always provide detailed contextual information about the study population.

Despite these limitations, the systematic review provided some useful insights into the differential impact of the COVID-19 pandemic on asylum seekers and undocumented migrants and can contribute to current and future pandemic response measures. 

## 5. Conclusions

Our review demonstrated that the COVID-19 pandemic has exacerbated pre-pandemic inequities between marginalised populations (asylum seekers and undocumented migrants) and the general population. The response specific to asylum seekers and undocumented migrants was characterised by pre-pandemic social and economic marginalisation, exclusion from pandemic induced policy measures, lack of appropriate pandemic communication, and reduced trust in governments and authority. Policies that reduce exposure and vulnerability to COVID-19 infection, grant equitable access to healthcare and social support, provide tailored messaging, and build capacities and resilience will help enable asylum seekers and undocumented migrants to cope with and recover from the pandemic shock. 

## Figures and Tables

**Figure 1 ijerph-19-06624-f001:**
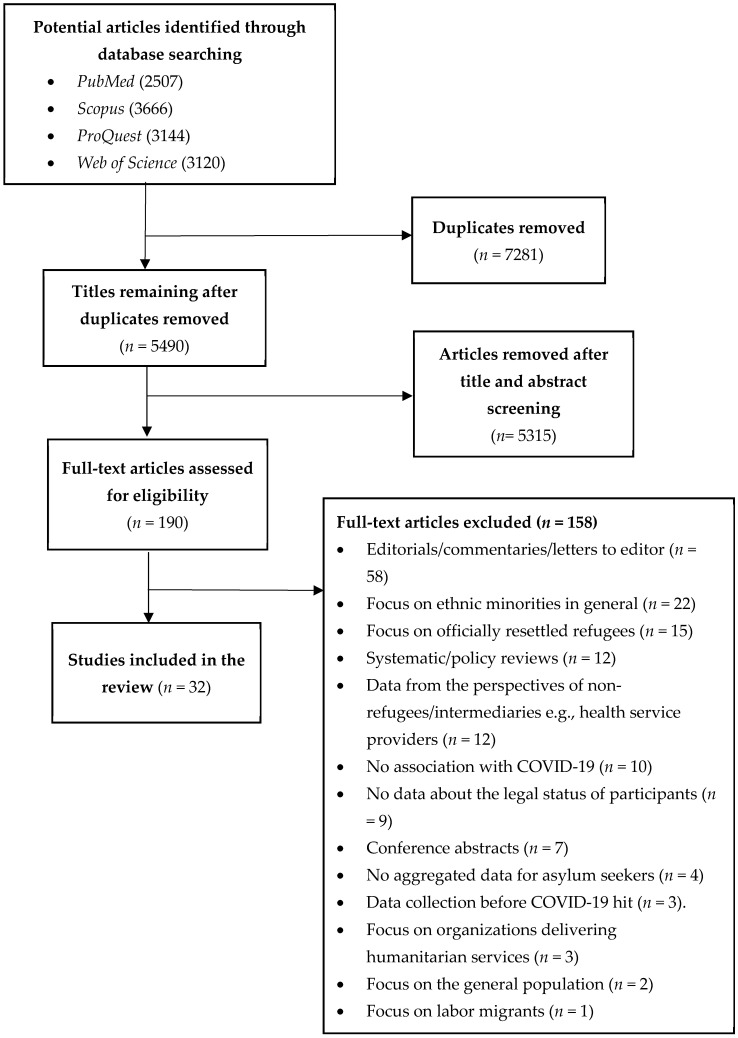
PRISMA flow chart of scoping review process and sampling.

**Table 1 ijerph-19-06624-t001:** Search terms and their combinations.

**COVID-19**	(TITLE-ABS-KEY (“2019 novel coronavirus disease” OR “ COVID-19 virus disease” OR “COVID 19” OR “ COVID-19 pandemic” OR “SARS-CoV-2 infection” OR “nCoV” OR “2019-nCoV disease” OR “2019-nCoV” OR “Novel corona” OR “novel-Covid” OR “ COVID-19” OR “2019 novel coronavirus infection” OR “ COVID-19 infection” OR “2019-nCoV infection” OR “coronavirus disease 2019” OR “coronavirus disease-19” OR “Coronavirus disease 2019” OR “ COVID-19 virus infection” OR “Covid” OR “covid-19” OR “COVID19” OR “Covid-2019” OR “Covid 2019” OR “covid-2019” OR corona OR “corona virus” OR Sars-Cov-2)
**Population**	AND TITLE-ABS-KEY (refugee OR “asylum seeker” OR “asyl*” OR (displaced AND (person* OR people)) OR “migra*” OR “forced migra*” OR migration OR immigra* OR “human migration” OR stateless OR “state-less” OR “irregular migra*” OR “regular migra*” OR “undocumented migra*” OR “internally displaced” OR “detainees” OR “residence status” OR “foreign-born” OR “displaced person” OR “noncitizen” OR “outsider” OR “newcomer” OR “newly arrived” OR “new arrival” OR “recent entrant” OR “non national” OR “non-national” OR “transient” OR “minorities” OR “ethnic”))
**Limits**	AND (LIMIT-TO (PUBYEAR, 2021) OR LIMIT-TO (PUBYEAR, 2021) OR LIMIT-TO (PUBYEAR, 2020) OR LIMIT-TO (PUBYEAR, 2019)) AND (LIMIT-TO (DOCTYPE, “ar”)) AND (LIMIT-TO (LANGUAGE, “English”))

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
