# Peer review of "“Lived the Pandemic Twice”: A Scoping Review of the Unequal Impact of the COVID-19 Pandemic on Asylum Seekers and Undocumented Migrants"

_ijerph, 2022, doi:10.3390/ijerph19116624_

Round 1
Reviewer 1 Report
Summary
The manuscript contributes a systematic review and narrative synthesis of the differential impact of the covid-19 pandemic on two related groups of marginalised people: asylum seekers and undocumented migrants. The main added value of the review over the articles it summarises is the use of Katikideddi and colleagues recently published mechanism-based framework for understanding inequalities. Overall the manuscript constitutes an important survey of the mechanisms that have contributed to differential impacts on two of the most marginalised groups across global society.
Detailed points:
* The summary in Table 2 would be strengthened considerably by including confidence intervals (or other measures of uncertainty and/or indications of statistical significance) around the findings quoted for quantitative studies. For improved readability, it would also be helpful in Table 2 to indicate which Katikireddi pathway(s) is/are captured by each study.
* Some parts of the narrative synthesis discuss before-after effects rather than differential effects, e.g. sentences 2 to 4 in Section 3.9. To avoid confusion, it would be better to separate these more clearly and also to focus the review on the differential effects, which is the main aim of the manuscript.
* In the study limitations, it should be stated more clearly that meta-analysis of the individual study findings has not been done (rather than just saying that calculating pooled measures was difficult, which is less clear). "Few limitations" should probably also read "some limitations".
* The structuring of the results in Section 3 is rather flat, meaning that six pathways of the framework do not appear prominently. If possible, it would be beneficial to use a further layer of sub-sectioning to pick out the different themes within each pathway.
* The review mostly reads very well. The sentences "This is critical not just for the lives of these already disadvantaged people. Exclusion of asylum seekers..." should probably be separated by a semi-colon rather than a full stop.
Author Response
Thank you for reviewing our paper and providing feedback that contributed to the improvement of the manuscript.

Reviewer 2 Report
The text is timely and well-founded. However, since most of the conclusions are derived from previous work, a greater precision of the recommendations for the future is lacking. The question that the authors must answer in the text is the following: what concrete policy measures could reduce exposure and vulnerability to a new pandemic and build capacity and resilience among asylum seekers and undocumented immigrants? Without the answer to this question, the text would be a good summary, but a summary after all of the previously published works.
Author Response

(The authors gave the same response as above.)

Reviewer 3 Report
The scoping review covered two important and timely topics with a comprehensive framework to examine the unparalleled impacts of COVID-19 on asylum seekers and undocumented migrants. It was nicely written and the focus was clearly presented. The inclusion of quotes from the individual study participants added valuable qualitative insights that are often missing in systematic reviews. Below are specific comments/suggestions.
Note: please include line numbers so it would be easier to identify issue locations for comments.
Abstract
Second sentence of Results: should be "greater exposure to the COVID-19 virus and higher infection rates.”
Introduction
The sentence with citation 10 or after citation 13: Please briefly explain what “community-based asylum seekers” are (how they are different from “asylum seekers” – the term you used more often in the paper).
Methods
Subsection 2.3. – the 6 pathways were already introduced in the Introduction. Therefore, to avoid repetition, the first sentence of the paragraph could be shortened or rephrased.
Explain how “high-income” country was defined.
Results
Section 3.1 – I would suspect it would be more accurate to add an descriptive word like “many” at the beginning of sentence 3 since not all asylum seekers or migrants continued to work during the pandemic.
Section 3.2 – It seems extreme that “all undocumented migrants had chronic comorbidities…”. It would be helpful to add information about the participant age or other demographics/characteristics of the cited study #36 as well as #52.
In addition, “vulnerability to infection” should not be limited to comorbidity. Perhaps identify additional factors, too. Similarly, Section 3.4 – The consequences identified in this section are more about finance and economic than social. Please also include some social elements as results of the pandemic or infection. (I wonder if it would work to merge section 3.14.)
To better mirror the framework adopted, consider labeling subsections 3.6 and 3.6 as 3.5.1 and 3.5.2 that you further developed (these are informative and well discussed). Otherwise, it appears that there is no study discussed in 3.5. Similarly, subsection 3.9 and 3.10 (maybe 3.11 and 3.12, too?) could be labeled under 3.8.
Regrouping these subsections could also improve the organization of the Results; 14 subsections are a lot and hard for readers to grasp the key findings.
Discussion
The integration of the citations in paragraphs 2 & 3 makes the presentation seem more like a literature review rather than discussing your study (review) results. Try to better distinguish what are your thoughts/recommendations and what are other researchers’ opinions or findings. The Discussion should offer more insights and implications; currently it’s more a summary than analytical discussion. In addition to recapturing the commonalities of the reviewed studies, also highlight some differences (e.g., dissimilarities among countries due to immigration policies, size of asylum seekers/migrants, or pandemic control or severity, etc.).
Last sentence of page 14, there is typo – should be “peak” of the pandemic, not pick.
Page 15 -- The first sentence of the first paragraph and the second sentence of the second paragraph are repetitive; the last sentence of the second paragraph is also quite similar. Please rephrase or shorten/delete some of them.
Conducting a systematic review is a big undertaking and the authors overall did a good job. Some revisions, additions, and slight reorganization could improve the value of the paper.
Author Response

(The authors gave the same response as above.)

Round 2
Reviewer 2 Report
The article has been properly revised
Author Response
Thank you for re-reviewing our paper.
Reviewer 3 Report
As I stated in the first review, conducting a systematic review is a big undertaking. The authors did a nice job organizing and synthesizing the included studies. The first 3 sections of the manuscript are in good shape. However, the revision seems rushed and the updated Discussion is not necessarily better than the original version (though I hate to say this!).
The three new paragraphs of the Discussion were written in a more casual language and less organized (e.g., the 3rd paragraph jumped from human rights to crowded housing to testing), not up to the standard of the rest of the manuscript, which is clear and must better written. The content is alright but the presentation style needs to be improved. Keep in mind that this is still a research paper and not a commentary; the authors need to take some time to carefully rephrase/revise/reorganize this section before the paper could be accepted.
For example, in particular (and not limited to just this point), saying “indeed and besides advocacy for systemic change” is inappropriate and also sounds awkward for a systematic review.
In the second paragraph of the Discussion, the half sentence “they are the results of policy decisions” should not be italic for emphasis since this is not an opinion piece. In the following sentence, “access to health” should be “access to healthcare.”
The last sentence “With adequate public support, funding and capacity-building, migrant-led organisations and specialized and community services can bridge the gap identified here and increase the general effectiveness of control measures, another important mechanism driving in-equitable COVID-19 harm according to Katikireddi et al’s” is confusing and probably miswritten. Why were these organizations and services a mechanism driving inequitable harm?
The addition of Pathway identification in the big summary table is not really helpful because some papers addressed more than one Pathway and you only listed one (inconsistent to the text), which could actually be misleading or confusing. Either remove the identification or list all the pathways mentioned in the text.
Further, one clarification is still needed. The 3rd sentence of section 3.2 “one record-based study report that all undocumented migrants had chronic comorbidities that worsened the severity and outcome of the infection” is likely inaccurate. Not all undocumented migrants had chronic comorbidities. The authors should consider adding a conditional clause like “who were hospitalized” or “who sought medical care” before “had chronic comorbidities” to correctly describe the study findings. (For reference, the following sentence is fine – it says “all undocumented migrants who died of COVID-19 in the US had at least one chronic disease condition.”)
A side note: the typo “pick” was deleted rather than corrected, which is okay but should be stated so in the response. Some of the suggestions for the Discussion in the first review was not addressed.
In summary, this is an important topic and the paper has great potential to contribute to the filed of knowledge. Understanding the revision time is limited, the paper still needs more work to make it publishable.
Author Response
Thank you again for re-reviewing our paper. We have accepted all your suggestions and revised the manuscript accordingly.
